# The Association between Sugar-Sweetened Beverages and High-Energy Diets and Academic Performance in Junior School Students

**DOI:** 10.3390/nu14173577

**Published:** 2022-08-30

**Authors:** Yan Ren, Chan Peng, Yanming Li, Feng Zhou, Mei Yang, Bing Xiang, Liping Hao, Xuefeng Yang, Jing Zeng

**Affiliations:** 1School of Public Health, Wuhan University of Science and Technology, Wuhan 430065, China; 2Research Center for Health Promotion in Women, Youth and Children, Wuhan University of Science and Technology, Wuhan 430065, China; 3Department of Nutrition and Food Hygiene, Tongji Medical College, Huazhong University of Science and Technology, Wuhan 430030, China

**Keywords:** junior school students, sugar-sweetened beverages, high-energy diets, academic performance, CEPS

## Abstract

This study aimed to understand the consumption frequency of sugar-sweetened beverages (SSBs) and high-energy diets in junior school students in China and to explore the relationship between SSBs and high-energy diets and academic performance. Information about 9251 junior school students was retrieved from the China Education Panel Survey (CEPS) database. The Mann–Whitney U test and the Kruskal–Wallis test were used to compare differences in academic performance based on the variables of interest. Generalized linear mixed models were used to analyze the association between the consumption frequency of SSBs and high-energy diet and student academic performance, fixed and random effects were included to control for confounding factors. The proportions of the “often” consumption group of SSBs and high-energy diets were 21.5% and 14.6%, respectively. For SSBs, the total score of the “often” consume group was 4.902 (95%CI: −7.660~−2.144, *p* < 0.001) points lower than that of the “seldom” consume group. Scores of Chinese math, and English were 0.864 (95%CI: −1.551~−0.177, *p* = 0.014), 2.164 (95%CI: −3.498~−0.831, *p* = 0.001), and 1.836 (95%CI: −2.961~−0.710, *p* = 0.001) points lower, respectively. For high-energy diets, the scores of total, Chinese and English in the “sometimes” consume group were 2.519 (95%CI: 0.452~4.585, *p* = 0.017), 1.025 (95%CI: 0.510~1.540, *p* < 0.001) and 1.010 (95%CI: 0.167~1.853, *p* = 0.019) points higher than that of the “seldom” consume group, respectively. Our findings suggested that consumption of SSBs was often negatively associated with academic performance in junior school students, while medium consumption of high-energy diets had a positive correlation. The positive association between high-energy diets and academic performance may be related to the food items included in the high-energy diets consumed by Chinese students. Schools and families should pay more effort to reduce the consumption of SSBs, and for high-energy diets, the focus should be on food selection and avoiding excessive intake. Longitudinal studies are needed to further test these findings among adolescents.

## 1. Introduction

The academic performance of junior school students is an important predictor of future success [1]. Academic achievement consists of performance metrics obtained from various academic subjects and itself reflects the acquisition of important knowledge and skills [2,3]. Academic performance is related to several variables, including unmodifiable factors such as heredity and gender [4] and modifiable factors such as family environment and personal habits [5]. In addition, parental education history also plays an important role [6]. Diet is another important factor; multiple studies have shown that poor dietary behavior, such as irregular breakfast consumption [7,8], and fast food intake [9] negatively impact academic performance and that some nutrients, such as iron [10,11], B group vitamins [12], and omega 3 [11], are positively correlated with academic performance. Compared with family environment and personal habits, dietary factors are of high concern, because childhood and adolescence are critical periods of physical, mental, and cognitive development, all of which are closely linked to nutritional status [13].

Among dietary factors that can influence academic performance, increasing attention has been paid to the effects of SSBs and high-energy diets. Studies have shown that, due to the gradual westernization of dietary patterns, the consumption frequency of SSBs by junior school students in China has been gradually increasing [14,15]. In addition, the intake of fried and other high-energy foods has also been increasing among junior school students. While offering a source of high caloric energy, such diets fail to provide adequate nutrients [16]. SSBs have been found to be harmful to health and their regular intake is strongly correlated with obesity [17], type 2 diabetes [18], and metabolic syndrome [19]. A high-energy diet increases the risk of cardiovascular [20] and metabolic diseases and can lead to high cholesterol and coronary heart disease [21].

Asides from their association with chronic diseases, poor dietary habits can also negatively affect academic performance. Previous studies have shown that lower intake of SSBs [22,23,24,25,26] and high-energy foods [27], especially fast food [26,27,28,29,30], is associated with higher academic achievement. Ickovics et al. found that children in fifth and sixth grades with low consumption of SSBs (<2 times per week) were more likely to pass the standardized test [22] and others have reported that students with higher academic performance consume fewer soda drinks [25]. Li et al. found that one unit of increase in the intake of high-energy food was associated with a decrease of 2.6 points (out of 100) in mathematics and 2.87 points (out of 100) in reading [27]. Other studies have referred to SSBs and high-energy diets more broadly as “junk food” (23); a study from Iceland reported a negative correlation between such “junk food” and a composite academic achievement score including multiple subjects among adolescents (*r* = −0.14 to −0.15) [29]. The mechanisms that can explain the associations already found in previous studies were mostly about the effects of students’ memory and concentration, which can lead to lower academic performance. However, it has also been shown that any fast food consumption may lead to a slight increase in academic performance from fifth and sixth grades, although the slowest growth was seen among children who reported eating fast food daily [31].

Based on these data, we believe that further research is required to improve our understanding of how poor eating habits affect academic performance in junior school students [32]. At present, this research field is still in a developmental stage. Most studies utilize self-reports to quantify academic performance and there is a lack of studies that use more objective approaches to quantify outcomes [23]. Furthermore, given the differences in economic development level and food system, the results of the impact may vary around the world [33]. However, there is still little data specifically addressing Asia, especially the Chinese situation and studies that can be generalized across the whole population [23].

Therefore, we aimed to investigate the relationship between diet (SSBs intake and high-energy foods) and academic performance in junior school students. We utilized the nationally-representative China Education Panel Survey (CEPS) database for source data. The academic performance of junior school students was evaluated via test scores, after controlling confounding factors, and the association between intake levels of SSBs and high-energy diets and academic performance was analyzed.

## 2. Materials and Methods

### 2.1. Study Participants and Sample Collection

We retrieved information about 9251 junior school students aged from 12–17 years from the CEPS database. CEPS is maintained by the National Survey Research Center at Renmin University of China and is a nationally representative, longitudinal, social survey that was established to investigate the impact of home, school, and community variables on individual educational outcomes. A stratified and multistage sampling design with probability proportional to size (PPS) was used in this survey. Firstly, 28 counties or districts were selected as primary sampling units. Secondly, four schools were randomly chosen within each selected county. Thirdly, within each school, we randomly selected two classes from the seventh grade and two classes from the ninth grade. Finally, the data from all students, parents, and teachers in the selected classes were utilized in the final survey sample [34].

Baseline data were selected from a nationwide multi-stage sample in the academic year 2013–2014; this included data from 10,279 students in seventh grades. A follow-up survey was conducted between 2014 and 2015. For this study, 9449 seventh grade students who were successfully followed up were screened, and 198 (2.1%) students were excluded as basic information was missing or not applicable. Finally, data from 9251 eighth grade students were included. Student data were collected by distributing student questionnaires which applied directly to students, and students reported on their own under the guidance of investigators. Basic information was obtained from baseline survey data, while dependent variables and influencing factors were obtained from the second follow-up survey.

### 2.2. SSBs and High-Energy Diet Intake Assessment

The influencing factors included SSBs intake and high-energy diet consumption by the students, both of which were included in the second follow-up questionnaire. Individuals responded to “How often do you drink sugary drinks (such as bubble tea) or carbonated drinks (such as cola)?” with “never, seldom, sometimes, often, and always”. The numbers of individuals who reported either never drinking SSBs or always drinking SSBs were very small; as such, we decided to combine these with their adjacent categories, thereby creating three response groups: seldom, sometimes, and often. The survey question on the frequency of consumption of high-energy diets was “How often do you eat fried, grilled, puffed, western fast food?” Responses were the same as for SSBs, with the same three categories being ultimately utilized.

### 2.3. Fixed and Random Effects

To model the relationship between SSBs and high-energy diets and academic performance, we included several potential confounders. Samples were obtained through multi-stage sampling. Twelve variables including student and family characteristics and personal habits were included in the study as fixed effects. Counties, schools, and classes were included as random effects. Student characteristics included gender, household registration, and living on campus. Family characteristics included household income, the highest level of parental education, whether there is a separate desk for study at home, whether there is a library at home, and home availability of internet and personal computer. Personal habits included time spent watching TV, playing games online, sleeping, and daily exercise.

### 2.4. Academic Performance

Academic performance was the main outcome variable. The academic performance of junior school students was deemed to be reflected by scores in core subjects such as Chinese, mathematics, and English. The total scores of these subjects were calculated to evaluate student performance. The original scores of the mid-term exams of 2014–2015 in these three subjects were available in the CEPS database. Since the full marks of the three subjects varied in different regions, we standardized them using a percentage scoring system and added them together to obtain a standardized total score. In addition, because test content and difficulty varied between regions, the counties, schools, and classes were also included in the analysis.

### 2.5. Statistical Analysis

After data collection, we first generated descriptive statistics. These included the general characteristics of the 9251 included junior school students and their consumption frequency of SSBs and a high-energy diet. Medians and quartiles were used to describe student scores because they did not follow a normal distribution. The independent effects of different variables on academic performance were also analyzed, the Mann–Whitney U test was used to compare differences in academic performance between the two classification variables, and the Kruskal–Wallis test (K–W test) was used to compare differences in academic performance between multiple classification variables. Since this study adopted the PPS sampling design, and the subjects were relatively simple and the ages were relatively concentrated, it was easier to obtain samples consistent with the population distribution. Therefore, no sample weighting was performed in our analysis.

Finally, we constructed three models using the generalized linear mixed model (GLMM) approach. We focused on how the consumption frequency of SSBs and high-energy diets influenced the academic performance of the cohort. Model I is an unadjusted model, the independent variables only included SSBs and high-energy meals. In Model II, student characteristics (gender, household registration, and living on campus), family characteristics (household income, highest level of parental education, whether there is a separate desk for study at home, whether there is a library at home, and availability of internet and computer) and personal habits (time spent watching TV, playing games online, sleeping, and daily exercise) were included into the GLMM as fixed effects. The results of Model I and Model II are shown in Appendix A. In Model III, apart from the variables included in Model II, we additionally adjusted for counties, schools, and classes; these were included as random effects. The Sidak method was used to correct for multiple testing. All analyses were carried out using IBM SPSS 26.0. A *p*-value of <0.05 was considered statistically significant.

## 3. Results

### 3.1. Participant Characteristics

Participant characteristics are shown in Table 1. Approximately half the students were male (52.0%, *n* = 4807) and had a rural household registration (hukou) (52.2%, *n* = 4833). Most students resided on campus (69.9%, *n* = 6471). The socioeconomic level was concentrated at the medium level (73.3%, *n* = 6777). Overall, 72.0% (*n* = 6657) of the highest achieved education level of the parents was high school and 20.1% (*n* = 1859) was college or above. Most reported having access at home to an independent study desk (78.6%, *n* = 7270), books (73.1%, *n* = 6768), and internet or computer (73.4%, *n* = 6792). Most students watched TV for less than 2 h per day (79.9%, *n* = 7389), played online games for less than 1 h per day (64.8%, *n* = 5998), slept 5–9 h per day (85.1%, *n* = 7876), and exercised less than 1 h per day (88.2%, *n* = 8160).

### 3.2. SSBs and High-Energy Diet Consumption

Figure 1 shows that for SSBs and high-energy diets, the frequency of the “seldom” group and the “sometimes” group is similar, both around 40%, and the lowest frequency for them was “often” group, 21.5% (*n* = 1989) for SSBs and 14.6% (*n* = 1345) for high-energy diet. There were statistically significant differences in the frequency of SSBs and high-energy diets between genders. Specifically, the proportion of male students who “often” consumed SSBs was higher than that of female students (23.7% vs. 19.1%, χ^2^ = 30.039, *p* < 0.001), and the proportion of female students who “often” consumed high-energy diets was higher than that of male students (16.0% vs. 13.4%, χ^2^ = 36.807, *p* < 0.001).

### 3.3. Distribution of Student Scores

Since the full marks of the three subjects varied in different regions, we standardized them using a percentage scoring system, and added them together to obtain a standardized total score, as shown in Table 2. Four dependent variables were obtained: the standardized total score, standardized Chinese score, standardized math score, and standardized English score. The standardized total score was used to evaluate the balanced developmental level of students and the individual subject scores reflect the development level of students in different aspects.

### 3.4. Univariate Analysis of Influencing Factors of Academic Performance

Results are shown in Table 3. For the different SSBs consumption frequencies, there was no significant difference in Chinese scores, but there were significant differences in total, math, and English scores. Higher SSBs consumption was associated with worse math scores. There were also significant differences in the three subject scores based on high-energy diet consumption. Students who sometimes ate a high-energy diet had the highest grades, followed closely by “seldom”; students who often ate a high-energy diet had the lowest scores.

In terms of student characteristics, there were differences in academic performance across all variables. Better grades were seen in female students, those with a non-rural hukou, and higher socioeconomic levels. The Chinese scores of the students who live on campus were better than those of the students who do not live on campus, while the results of total score, mathematics, and English scores were the opposite. Family characteristics were also associated with performance; students with more highly-educated parents, access to a separate desk, a large collection of books at home, and access to a computer or internet had higher scores. In terms of personal habits, we found better academic performance in students who reported moderate sleeping time, watching less TV, and playing fewer games. Somewhat surprisingly, we also found that less exercise was associated with higher academic performance, but this result may also be due to improper grouping, for most subjects (88.2%) were within the category of “≤60 min”.

### 3.5. GLMM Analysis of SSBs and High-Energy Diet and Academic Performance

After identifying significant differences between different consumption frequencies of SSBs and high-energy diet and academic performance, we performed additional analyses. Since the scores were not normally distributed and the research objects in this paper were obtained through multi-stage sampling, we used a generalized linear mixed model to analyze the association of SSBs and high-energy diets with academic performance. Three models were established. Model I (Appendix A) showed the association of SSBs and high-energy diets with academic performance without adjusting for relevant variables. In Model II (Appendix A), fixed effects were added to Model I, including student characteristics, family characteristics, and personal habit characteristics. Model III (Table 4) examined the effects of study factors on academic performance after controlling for random effects (county, school, and class).

After controlling for fixed and random effects, the total score of the high intake group was 4.902 (95%CI: −7.660~−2.144, *p* < 0.001) points lower than that of the low intake group (Table 4). In addition, we can see a positive correlation between the medium high-energy diets and academic performance in all models. After controlling for other factors, the average total score was 2.519 (95%CI: 0.452~4.585, *p* = 0.017) points higher in the moderate intake group than in the low intake group (Table 4).

For Chinese scores, after controlling for random effects (Table 4), we found that high-frequency SSBs intake negatively affected Chinese scores compared with low-level intake. The “often” group was 0.864 (95%CI: −1.551~−0.177, *p* = 0.014) points lower than the “seldom” group. Similarly, we can see a positive effect of the medium high-energy diets on academic performance in all models. After controlling for other factors, the average Chinese score was 1.025 (95%CI: 0.510~1.540, *p* < 0.001) points higher in the moderate intake group than in the low intake group (Table 4).

The negative effect of SSBs on mathematics is shown in Table 4. After controlling for other variables, the “often” group was 2.164 (95%CI: −3.498~−0.831, *p* = 0.001) points lower than the “seldom” group. We also found that high-frequency consumption of a high-energy diet led to lower math scores in Models I and II (Appendix A), however, this effect was not observed in Models III.

The impact on English scores was highly consistent in the three models, which is shown in Table 4 and Appendix A. We found that high SSBs consumption frequency led to lower English scores, and medium frequency consumption of high-energy diet led to higher English scores, whether this was controlled for other effects or not. After controlling for fixed and random effects, for SSBs, the “often” group was 1.836 (95%CI: −2.961~−0.710, *p* = 0.001) points lower than the “seldom” group, while for high-energy diet, the “sometimes” group was 1.010 (95%CI: 0.167~1.853, *p* = 0.019) points higher than the “seldom” group.

## 4. Discussion

In the current study, we found a relatively high proportion of the students in “often” consumption groups of SSBs and high-energy diets. We also found an association between SSBs and high-energy diets and test scores. Our univariate analyses indicated that students who reported higher consumption of SSBs had poorer test scores, while those who reported medium consumption of high-energy diet had better scores. GLMM analyses showed that, after controlling for fixed and random effects, both SSBs and high-energy diets were still associated with scores. Specifically, high frequency of SSBs consumption had a negative statistically significant effect on total scores and all three subjects, while moderate consumption of high-energy diets had a positive statistically significant effect on total scores, Chines scores and English scores. Our study suggests that dietary modification support for junior school students is likely to benefit health and education outcomes.

Due to the development of the food industry, the fast food industry and people’s fast food consumption has grown rapidly globally, especially among children and adolescents in low- and middle-income countries [35]. In China, based on the CHNS project, the proportion of children and adolescents consuming SSBs (≥2 times/week) increased from 14.2% in 2004 to 21.8% in 2011 [36]. A survey based on primary and middle school students in Nanjing found that the proportions of consuming SSBs and fast food (≥1 time/week) were 30.4% and 30.2%, respectively [37]. Combined with the results of this study, the proportions of “often” group of SSBs and high-energy diets were 21.5% and 14.6%, respectively. It can be seen that the consumption of SSBs and high-energy diets by Chinese children and adolescents is increasing rapidly, which is worrying.

A large body of evidence suggests that a healthy diet during childhood and adolescence helps prevent cardiovascular and other chronic diseases in the short, medium, and long term [38]. However now, the focus has extended to investigating the effects of poor diet on academic performance; the relationship between eating behavior and academic achievement has received some support [23,39]. Previous studies have classified SSBs and high-energy diets as two critical variables in a “junk food” category [23]. A negative influence has been reported between “junk food” and academic performance in math, English, and science [30]. A Norwegian study showed that a diet rich in unhealthy foods (such as soft drinks, sweets, chocolate, chips, and fast food) was associated with math learning difficulties [40].

In our study, we found that a high SSBs consumption frequency was associated with lower total and component test scores; these findings are consistent with prior research [41,42,43]. Based on a cross-sectional study of middle school students in Aragon, Spain, it was found that the consumption of sweets and SSBs is inversely related to average academic scores (*r* = −0.205) [41]. Studies of Australian school-age children have also reported that increased consumption of SSBs is associated with significantly lower test scores in reading, writing, grammar/punctuation, and numeracy [13]. A study of teenagers from Chongqing, China, has also shown that high-frequency consumption of SSBs is associated with poorer academic grades [44]. By analyzing the literature, we found that there were currently no studies that have found a positive correlation between SSBs and academic performance. Therefore, it can be considered that it is necessary for junior school students to reduce their intake of sugary drinks.

From the available evidence, the effect of high-energy diets on academic performance is controversial. Most studies have shown that lower intakes of fast food are associated with higher academic achievement [23,27,28,29,30]. Conversely, Purtell et al. reported that any fast food consumption was associated with small gains in academic growth when kindergarten children were followed up at eighth grade [31]. A study of primary school students in one province of Canada showed that although excessive intake of saturated fatty acids affected reading and writing proficiency, it did not affect math performance [45]. Similarly, after adjusting for multiple covariates, we also found that moderate-consumption frequency of high-energy food improved the level of total scores, Chinese, and English. This might be because our cohort included a different age group or because of specific dietary factors (e.g., the high-energy diet in our study may differed from those consumed by cohorts in other studies). In western countries, a high-energy diet is generally characterized by high fat and sugar [31], but in China, a high-energy diet may be high in protein in addition to high fat and carbohydrate. For example, in China, most of the grilled food is meat, but also soy products, which are high in protein, and high protein has a positive effect on cognition [46]. Moreover, the consumption of high-energy diets in developing countries is often associated with better economic conditions, in our study; the bivariate correlation analysis showed that there was a positive correlation between the consumption frequency of high-energy diet and the household income (spearman correlation, *r* = 1.070, *p* < 0.001), which can also positively affect cognitive performance [47].

Although most research has focused on the relationship between nutrition and cognitive development in children, the mechanism by which SSBs and high-energy diets can affect academic performance is not well understood. Regarding SSBs, this may be because blood glucose concentration can affect concentration; a meta-analysis of breakfast with different glycemic index or glycemic load showed that higher postprandial glucose responses harmed mental performance in children, which would lead to lower academic performance [48]. Additionally, there is evidence that there are deleterious relations between sugar and other simple carbohydrate consumptions and school children’s memory and attention processes [49]. Another study indicated that high sugar foods and additives can influence attention and learning behavior [50]. So, these may partly explain why the high-frequency intake of SSBs is associated with lower academic performance. In terms of high-energy diets, I have previously explained why our findings differ from those of other studies. However, we also saw that in Models I and II in Appendix A, the “often consume” group had lower math scores compared to the “seldom consume” group. Therefore, we can hypothesize that the relationship between excessive intake of high-energy diets and academic performance should still be negative, because a prominent feature of high-energy meals is high fat, while studies have found that high-fat meals are associated with longer reaction times [51], poorer concentration and processing speed [50], there are also studies showing that it is negatively correlated with working memory [52] and cognition [46]. In addition, high-energy diets are associated with obesity and metabolic diseases [53], which are associated with cognitive dysfunction and a higher risk of developing neurodegenerative diseases such as Alzheimer Disease [54,55]. Therefore, even though a “moderate” intake of high-energy diets in students is associated with better outcomes in academic performance, we should not encourage the consumption of high-energy diets arbitrarily because it may lead to increased consumption of those foods that may not be beneficial for health in the long run. Due to the complexity of high-energy foods, we believe the focus should be on food choices and avoiding excessive intake.

We used a large, nationally representative database for our data source. We utilized GLMM modeling to analyze total academic and subject scores. The results of this study should inform educators and school administrators about the importance of diet. We believe that parents, educators, and welfare departments should set school standards to ensure that students are obtaining a healthy diet. In particular, there should be an effort to reduce consumption of SSBs and meanwhile control the excessive consumption of high-energy diets.

### Strengths and Limitations

Our study has the following advantages. Firstly, the sample is a representative of the Chinese school population and therefore widely generalizable. Secondly, since the variables were not independent and scores did not follow a normal distribution, GLMM was used to analyze the relationship between SSBs and high-energy diets and academic performance. This model is more in line with the actual situation of the data and is considered to be an effective method to explore the real relationship between variables in the real world.” Thirdly, student academic performance was assessed based on objective test scores rather than self-reports. There were some limitations. Firstly, as a cross-sectional study, we could not establish a clear causal relationship between the study variables. The reporting period for the dietary data was not specified which made it difficult to determine whether the reporting period for dietary data coincided with the reporting period for academic achievement. Furthermore, we only analyzed the dietary frequencies of SSBs and high-energy diet; no other dietary behaviors were investigated. The consumption frequency was self-reported and students’ judgment of frequency is subjective, so we need to admit the possibility of some reporting bias. In addition, the academic performance was only evaluated based on the core subjects of Chinese, mathematics, and English, and other subjects were not included, which may not fully reflect the students’ learning status.

## 5. Conclusions

Our findings suggest that SSBs are negatively associated with academic performance in junior school students, while medium high-energy diets have a positive relationship. Students reporting a higher consumption frequency of SSBs had lower school scores, and those who reported a medium consumption frequency of a high-energy diet had better scores. As academic performance and long-term outcomes are closely related, this study highlights the importance of dietary health education to promote a healthy lifestyle in school-aged children. Schools and families should pay more effort to reduce the consumption of SSBs, and for high-energy diets, the focus should be on food selection and avoiding excessive intake. Further longitudinal studies and randomized controlled trials are needed to assess the impact of SSBs and high-energy diets on youth academic outcome.

## Figures and Tables

**Figure 1 nutrients-14-03577-f001:**
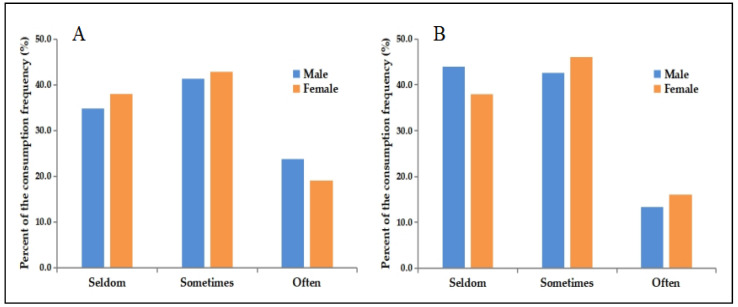
The percent of the consumption frequency of SSBs and high-energy diets in male students and female students. (**A**) The data represent SSB consumption frequency. (**B**) The data represent high-energy diet consumption frequency.

**Table 1 nutrients-14-03577-t001:** Description of the general participant characteristics.

	Variables		*N*	Proportion (%)	95%CI
Student Characteristics	Gender	Female	4444	48.0	(47.0, 49.1)
		Male	4807	52.0	(50.9, 53.0)
	Region of residence	Urban	4418	47.8	(46.7, 48.8)
		Rural	4833	52.2	(51.2, 53.3)
	Campus residence	No	2780	30.1	(29.1, 31.0)
		Yes	6471	69.9	(69.0, 70.9)
Family Characteristics	Household income	Low	1418	15.3	(14.6, 16.1)
		Medium	6777	73.3	(72.3, 74.2)
		High	1056	11.4	(10.8, 12.1)
	Highest level of Parental Education	Primary or below	735	7.9	(7.4, 8.5)
		High school	6657	72.0	(71.0, 72.9)
		College or above	1859	20.1	(19.3, 20.9)
	Independent desk	Yes	7270	78.6	(77.7, 79.4)
		No	1981	21.4	(20.6, 22.3)
	Domestic library	Lower	1135	12.3	(11.6, 12.9)
		Low	1348	14.6	(13.9, 15.3)
		Medium	3547	38.3	(37.4, 39.3)
		High	2095	22.6	(21.8, 23.5)
		Higher	1126	12.2	(11.5, 12.8)
	Internet and Computer	No	2459	26.6	(25.7, 27.5)
		One of them	6792	73.4	(72.5, 74.3)
Personal habits	Screen (TV)	No	2940	31.8	(30.8, 32.7)
		<1 h	2267	24.5	(23.6, 25.4)
		1–2 h	2182	23.6	(22.7, 24.5)
		2–3 h	1045	11.3	(10.7,12.0)
		3–4 h	312	3.4	(3.0, 3.8)
		>4 h	505	5.5	(5.0, 5.9)
	Play online games	No	3954	42.7	(41.7, 43.8)
		<1 h	2044	22.1	(21.3, 22.9)
		1–2 h	1684	18.2	(17.4, 19)
		2–3 h	779	8.4	(7.9, 9.0)
		3–4 h	289	3.1	(2.8, 3.5)
		>4 h	501	5.4	(5.0, 5.9)
	Sleep	<5 h	61	0.7	(0.5, 0.8)
		5–9 h	7876	85.1	(84.4, 85.9)
		>9 h	1314	14.2	(13.5, 14.9)
	Sports	≤60 min	8160	88.2	(87.5, 88.9)
		60–180 min	967	10.5	(9.8, 11.1)
Dietary factors		>180 min	124	1.3	(1.1, 1.6)
	SSBs	Seldom	3366	36.4	(35.4, 37.4)
		Sometimes	3896	42.1	(41.1, 43.1)
		Often	1989	21.5	(20.7, 22.3)
	High-energy diet	Seldom	3801	41.1	(40.1, 42.1)
		Sometimes	4096	44.3	(43.3, 45.3)
		Often	1345	14.6	(13.9, 15.4)

**Table 2 nutrients-14-03577-t002:** Description of the academic performance of these junior school students.

Variables	N	Male	Female	Total
		M (P25, P75) ^a^	M (P25, P75) ^a^	M (P25, P75) ^a^
Standardized total score	9251	192.5 (189.7, 195.0)	220.8 (218.3, 222.5)	206.7 (153.0, 242.5)
Standardized Chinese score	9251	68.0 (67.5, 69.0)	75.0 (75.0, 75.8)	71.7 (61.3, 79.0)
Standardized Math score	9251	67.0 (66.0, 68.7)	73.0 (72.3, 74.0)	70.0 (45.0, 85.4)
Standardized English score	9251	59.0 (57.5, 60.0)	73.3 (72.7, 74.5)	66.7 (43.0, 82.7)

^a^ Data were not normally distributed therefore medians and quartiles were used.

**Table 3 nutrients-14-03577-t003:** Univariate analysis of influencing factors of academic performance ^a^.

			Total Scores		Chinese Scores		Math Scores		English Scores	
Variables		N	Average Rank ^b^	*p*-Value	Average Rank ^b^	*p*-Value	Average Rank ^b^	*p*-Value	Average Rank ^b^	*p*-Value
SSBs	Seldom	3366	4670.1	<0.001	4605.2	0.068	4725.6	<0.001	4652.6	<0.001
	Sometimes	3896	4699.5		4693.7		4658.2		4713.6	
	Often	1989	4407.3		4528.5		4394.5		4409.5	
High-energy diet	Seldom	3801	4564.8	<0.001	4495.1	<0.001	4658.5	<0.001	4520.0	<0.001
	Sometimes	4096	4781.9		4789.5		4716.0		4804.1	
	Often	1354	4326.3		4498.8		4262.5		4384.9	
Gender	Female	4444	5199.5	<0.001	5320.8	<0.001	4887.6	<0.001	5304.7	<0.001
	Male	4807	4095.8		3983.7		4384.1		3998.6	
Region of residence	Urban	4418	5031.1	<0.001	4820.6	<0.001	4878.4	<0.001	5180.3	<0.001
	Rural	4833	4255.7		4448.1		4395.3		4119.3	
Campus Residence	No	2780	4375.3	<0.001	4739.6	0.007	4525.8	0.018	4145.6	<0.001
	Yes	6471	4733.7		4577.2		4669.1		4832.4	
Household Income	Low	1418	3496.4	<0.001	3776.7	<0.001	3733.9	<0.001	3403.6	<0.001
	Medium	6777	4756.9		4729.2		4740.6		4757.5	
	High	1056	5302.7		5104.4		5088.4		5423.8	
Parental Education	Primary or below	735	3155.5	<0.001	3205.3	<0.001	3433.8	<0.001	3256.2	<0.001
	High school	6657	4410.6		4532.4		4453.2		4360.4	
	College or above	1859	5978.9		5522.9		5716.3		6118.7	
Independent desk	Yes	7270	4937.4	<0.001	4833.1	<0.001	4852.8	<0.001	4991.3	<0.001
	No	1981	3483.2		3866.0		3793.7		3285.3	
Home collection	Lower	1135	3217.0	<0.001	3613.5	<0.001	3559.1	<0.001	3054.7	<0.001
	Low	1348	4058.0		4256.6		4322.1		3863.3	
	Medium	3547	4488.1		4514.9		4493.5		4499.9	
	High	2095	5486.2		5260.3		5216.3		5635.1	
	Higher	1126	5560.2		5258.8		5384.4		5642.6	
Internet and Computer	No	2459	3525.6	<0.001	3769.2	<0.001	3781.8	<0.001	3451.0	<0.001
	Only one	6792	5024.4		4936.2		4931.6		5051.4	
Screen time (TV)	No	2940	5296.7	<0.001	5184.8	<0.001	5264.9	<0.001	5217.1	<0.001
	<1 h	2267	4780.7		4685.6		4746.6		4836.6	
	1–2 h	2182	4289.9		4346.3		4304.6		4329.9	
	2–3 h	1045	3936.0		4097.9		3979.8		3976.3	
	3–4 h	312	4007.3		4209.9		4097.5		3961.3	
	>4 h	505	3289.4		3663.7		3417.5		3274.0	
Play online games	No	3954	4930.3	<0.001	4861.5	<0.001	4970.6	<0.001	4857.7	<0.001
	<1 h	2044	4988.9		4888.9		4905.1		5024.3	
	1–2 h	1684	4313.1		4409.5		4277.6		4374.0	
	2–3 h	779	4044.7		4178.1		4024.2		4153.3	
	3–4 h	289	3798.2		4057.2		3903.0		3758.6	
	>4 h	501	3176.7		3447.4		3291.9		3254.9	
Sleep	<5 h	61	2690.6	<0.001	2251.3	<0.001	2974.2	<0.001	3041.9	<0.001
	5–9 h	7876	4779.0		4742.7		4754.2		4783.6	
	>9 h	1314	3798.9		4036.7		3934.2		3754.7	
Sports	≤60 min	8160	4685.5	<0.001	4698.7	<0.001	4666.1	<0.001	4682.8	<0.001
	60–180 min	967	4280.5		4162.7		4422.1		4288.1	
	>180 min	124	3402.9		3456.2		3580.4		3522.5	

^a^ Binary-classification data in the table were analyzed by Mann–Whiney U test for comparison. Multi-classification data in the table were analyzed by Kruskal–Wallis test for comparison, *p* < 0.05 is statistically significant. ^b^ Average rank: since scores do not follow a normal distribution, rank is used to represent the relative positions of data. The average rank is the average of all ranks. The higher the average rank, the better the performance.

**Table 4 nutrients-14-03577-t004:** Association of SSBs and high-energy diet with total academic scores, Chinese, Math, and English scores from the adjusting GLMM analyses (Model III).

Subjects	Variables		*β*	95%CI	*p*-Value
Total scores	SSBs	Seldom (reference)	0.000	—	—
		Sometimes	−2.904	(−5.041, −0.767)	0.008
		Often	−4.902	(−7.660, −2.144)	<0.001
	High-energy diet	Seldom (reference)	0.000	—	—
		Sometimes	2.519	(0.452, 4.585)	0.017
		Often	−0.357	(−3.418, 2.704)	0.819
Chinese	SSBs	Seldom (reference)	0.000	—	—
		Sometimes	−0.531	(−1.063, 0.002)	0.051
		Often	−0.864	(−1.551, −0.177)	0.014
	High-energy diet	Seldom (reference)	0.000	—	—
		Sometimes	1.025	(0.510, 1.540)	<0.001
		Often	0.521	(−0.242, 1.284)	0.181
Math	SSBs	Seldom (reference)	0.000	—	—
		Sometimes	−1.567	(−2.600, −0.534)	0.003
		Often	−2.164	(−3.498, −0.831)	0.001
	High-energy diet	Seldom (reference)	0.000	—	—
		Sometimes	0.483	(−0.516, 1.482)	0.343
		Often	−1.105	(−2.585, 0.375)	0.143
English	SSBs	Seldom (reference)	0.000	—	—
		Sometimes	−0.848	(−1.720, 0.024)	0.057
		Often	−1.836	(−2.961, −0.710)	0.001
	High-energy diet	Seldom (reference)	0.000	—	—
		Sometimes	1.010	(0.167, 1.853)	0.019
		Often	0.214	(−1.035, 1.464)	0.737

The GLMM analyses were adjusted by fixed effects and random effects. Fixed effects (gender, household registration, on campus residence, socioeconomic level, parental education level, desk, book collection, internet and computer, TV watching time, internet time, sleep time, and exercise time); Random effects (county, school, and class). The *p*-values were adjusted for multiple comparisons.

## Data Availability

The database used in the study was available from the corresponding author upon reasonable request.

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
