# Peer review of "The Association between Sugar-Sweetened Beverages and High-Energy Diets and Academic Performance in Junior School Students"

_nutrients, 2022, doi:10.3390/nu14173577_

Round 1
Author Response
Response to Reviewer 1 Comments
We sincerely thank Reviewer 1 for your scholarly review and valuable suggestions on our present manuscript. We have responded in detail to each of the points raised and these responses are given below.
In this manuscript Peng et al. presented the results of a study based on a survey and aiming to evaluate the association between sugar-sweetened beverages or high-energy diets and academic performance in Chinese students. They found that students who answered “often” to the survey question regarding the consumption of sugar-sweetened beverages had a lower academic performance as compared to those who answered “seldom”. However, students having a high energy diet had a better performance. This last result is unexpected and requires some additional comments to help the reader in the interpretation of the results.
Please find below my specific comments:
Point 1: Abstract. The conclusion on high energy diet does not reflect the result.
Response 1: Thank you for your careful review and kind reminding. We have changed the sentence “Schools and families should pay more effort to reduce consumption of SSBs and meanwhile control the excessive consumption of high-energy diets.” into “Schools and families should pay more effort to reduce consumption of SSBs and meanwhile allowing students to eat moderate amounts of high-energy foods.” (Line 35-37)
Point 2: Material and methods, line 96. Please report the age range of the students.
Response 2: Thank you very much for your valuable advice. We have added the age range of the students in the Material and methods section. The sentence is as following: “We retrieved information about 9251 junior school students aged from 12-17 years from the CEPS database.” (Line 102)
Point 3: SSBs and High-energy Diet Intake Assessment, line 121. Please make clear which was the survey question used to obtain the information on the frequency of consumption of high-energy diets.
Response 3: Thank you for your precious and helpful suggestions to make our manuscript readable. We have changed the sentence ‘Individuals also responded to “How often do you eat fried, grilled, puffed, western fast food? ”’ into ‘The survey question on the frequency of consumption of high-energy diets was “How often do you eat fried, grilled, puffed, western fast food?” ’ (Line 129-130)
Point 4: Fixed and Random Effects, lines 123-133. I would integrate this part in the “Statistical analysis” subsection.
Response 4: Thank you very much for your beneficial advice. Because we mentioned these fixed and random effects in “Statistical analysis” when introduce the Generalized Linear Mixed Model, so if we integrated “Fixed and Random Effects” part into the “Statistical analysis” subsection, it's a little repetitive. Therefore, after weighing, we feel that it is better not to integrate it.
Point 5: Statistical analysis, lines 147-148. The sentence sound odd and should be rephrased, eg. “…because they did not follow a normal distribution”.
Response 5: Thank you for your careful review and rigorous academic attitude. We have changed the sentence “Medians and quartiles were used to describe student scores because they did not assume a normal distribution.” into “Medians and quartiles were used to describe student scores because they did not follow a normal distribution.” (Line 158)
Point 6: Statistical analysis, line 156. Replace “non-adjusting model” with “unadjusted model”.
Response 6: Thank you for your careful review and kind reminding. We have replaced “non-adjusting model” with “unadjusted model”. (Line 167)
Point 7: Statistical analysis, the complex design of the survey should be taken into account in the analysis. Therefore, the random effects should be included in all the models used in the analysis. Otherwise, the variance of the estimates is not correct.
Response 7: Thank you for your precious and helpful suggestions to make our manuscript more reasonable. As you said, the results of Model III are the results that we focus on analyzing and discussing. According to your advice, we combined the Tables 4-7 as Table 4 by reporting only the estimates obtained by Model III. Meanwhile, to illustrate the whole analysis process of the GLMM models, we keep the results of Model I and Model II as supplementary material (Table S1). Table 4 and Table S1 are as following.
Table 4. Association of SSBs and high-energy diet with total academic scores, Chinese, Math, and English scores from the adjusting GLMM Analyses (Model III).
|
Subjects |
Variables |
β |
95%CI |
p-value |
|
|
Total scores |
SSBs |
Seldom (reference) |
0.000 |
− |
− |
|
|
|
Sometimes |
−2.904 |
(−5.041, -0.767) |
0.008 |
|
|
|
Often |
−4.902 |
(−7.660,−2.144) |
<0.001 |
|
|
High-energy diet
|
Seldom (reference) |
0.000 |
− |
− |
|
|
Sometimes |
2.519 |
(0.452, 4.585) |
0.017 |
|
|
|
Often |
−0.357 |
(−3.418, 2.704) |
0.819 |
|
|
Chinese |
SSBs |
Seldom (reference) |
0.000 |
— |
— |
|
|
|
Sometimes |
−0.531 |
(−1.063, 0.002) |
0.051 |
|
|
|
Often |
−0.864 |
(−1.551,−0.177) |
0.014 |
|
|
High-energy diet
|
Seldom (reference) |
0.000 |
— |
— |
|
|
Sometimes |
1.025 |
(0.510, 1.540) |
<0.001 |
|
|
|
Often |
0.521 |
(−0.242,1.284) |
0.181 |
|
|
Math |
SSBs |
Seldom (reference) |
0.000 |
— |
— |
|
|
|
Sometimes |
−1.567 |
(−2.600, −0.534) |
0.003 |
|
|
|
Often |
−2.164 |
(−3.498,−0.831) |
0.001 |
|
|
High-energy diet
|
Seldom (reference) |
0.000 |
— |
— |
|
|
Sometimes |
0.483 |
(−0.516, 1.482) |
0.343 |
|
|
|
|
Often |
−1.105 |
(−2.585, 0.375) |
0.143 |
|
English |
SSBs |
Seldom (reference) |
0.000 |
— |
— |
|
|
|
Sometimes |
−0.848 |
(−1.720, 0.024) |
0.057 |
|
|
|
Often |
−1.836 |
(−2.961,−0.710) |
0.001 |
|
|
High-energy diet |
Seldom (reference) |
0.000 |
— |
— |
|
|
|
Sometimes |
1.010 |
(0.167, 1.853) |
0.019 |
|
|
|
Often |
0.214 |
(−1.035, 1.464) |
0.737 |
The GLMM analyses were adjusted by fixed effects and random effects. Fixed effects (gender, household registration, on campus residence, socioeconomic level, parental education level, desk, book collection, internet & computer, TV watching time, internet time, sleep time, and exercise time); Random effects (county, school, and class).
Table S1 Association of SSBs and high-energy diet with total academic scores, Chinese, Math, and English scores from the adjusting GLMM Analyses. (Model I and Model II)
|
Subjects |
Models |
Variables |
β |
95%CI |
p-value |
|
|
Total scores |
Model I a |
SSBs |
Seldom (reference) |
0.000 |
− |
− |
|
|
|
Sometimes |
−1.610 |
(−4.513, 1.293) |
0.277 |
|
|
|
|
|
Often |
−5.797 |
(−9.486,−2.108) |
0.002 |
|
|
High-energy diet |
Seldom (reference) |
0.000 |
− |
− |
|
|
|
Sometimes |
6.909 |
(4.107, 9.711) |
<0.001 |
||
|
|
Often |
−0.706 |
(−4.806,3.393) |
0.736 |
||
|
|
Model II b |
SSBs |
Seldom (reference) |
0.000 |
− |
− |
|
|
|
|
Sometimes |
−2.053 |
(−4.634, 0.528) |
0.119 |
|
|
|
|
Often |
−3.198 |
(−6.512,−0.116) |
0.059 |
|
|
High-energy diet |
Seldom (reference) |
0.000 |
− |
− |
|
|
|
Sometimes |
2.721 |
(0.224, 5.218) |
0.033 |
||
|
|
Often |
−2.145 |
(−5.812,1.522) |
0.252 |
||
|
Chinese |
Model I a |
SSBs |
Seldom (reference) |
0.000 |
− |
− |
|
|
|
|
Sometimes |
−1.610 |
(−4.513, 1.293) |
0.277 |
|
|
|
|
Often |
−5.797 |
(−9.486,−2.108) |
0.002 |
|
|
|
High-energy diet |
Seldom (reference) |
0.000 |
− |
− |
|
|
|
Sometimes |
6.909 |
(4.107, 9.711) |
<0.001 |
|
|
|
|
|
Often |
−0.706 |
(−4.806,3.393) |
0.736 |
|
|
Model II b |
SSBs |
Seldom (reference) |
0.000 |
− |
− |
|
|
|
|
Sometimes |
−2.053 |
(−4.634, 0.528) |
0.119 |
|
|
|
|
Often |
−3.198 |
(−6.512,−0.116) |
0.059 |
|
|
|
High-energy diet |
Seldom (reference) |
0.000 |
— |
— |
|
|
|
|
Sometimes |
1.245 |
(0.589, 1.901) |
<0.001 |
|
|
|
|
Often |
0.337 |
(−0.627, 1.300) |
0.493 |
|
Math |
Model I a |
SSBs |
Seldom (reference) |
0.000 |
— |
— |
|
|
|
|
Sometimes |
−0.731 |
(−2.014, 0.553) |
0.264 |
|
|
|
|
Often |
−2.064 |
(−3.695, −0.433) |
0.013 |
|
|
|
High-energy diet |
Seldom (reference) |
0.000 |
— |
— |
|
|
|
|
Sometimes |
1.312 |
(0.073, 2.551) |
0.038 |
|
|
|
|
Often |
−2.301 |
(−4.113,−0.488) |
0.013 |
|
|
Model II b |
SSBs |
Seldom (reference) |
0.000 |
— |
— |
|
|
|
|
Sometimes |
−0.959 |
(−2.156, 0.238) |
0.116 |
|
|
|
|
Often |
−1.360 |
(−2.897,0.177) |
0.083 |
|
|
|
High-energy diet |
Seldom (reference) |
0.000 |
— |
— |
|
|
|
|
Sometimes |
0.264 |
(−0.895, 1.422) |
0.656 |
|
|
|
|
Often |
−2.071 |
(−3.772,−0.370) |
0.017 |
|
English |
Model I a |
SSBs |
Seldom (reference) |
0.000 |
— |
— |
|
|
|
|
Sometimes |
−0.610 |
(−1.810, 0.591) |
0.319 |
|
|
|
|
Often |
−2.672 |
(−4.197,−1.147) |
0.001 |
|
|
|
High-energy diet |
Seldom (reference) |
0.000 |
— |
— |
|
|
|
|
Sometimes |
3.274 |
(2.116, 4.433) |
<0.001 |
|
|
|
|
Often |
0.438 |
(−1.257, 2.132) |
0.613 |
|
|
Model II b |
SSBs |
Seldom (reference) |
0.000 |
— |
— |
|
|
|
|
Sometimes |
−0.753 |
(−1.804, 0.299) |
0.160 |
|
|
|
|
Often |
−1.498 |
(−2.848,−0.148) |
0.030 |
|
|
|
High-energy diet |
Seldom (reference) |
0.000 |
— |
— |
|
|
|
|
Sometimes |
1.213 |
(0.196, 2.230) |
0.019 |
|
|
|
|
Often |
−0.411 |
(−1.905, 1.083) |
0.590 |
a Model I:Non-adjusted model (independent variables include SSBs and high-energy diets). b Model II: Model 1+ Fixed effects (gender, household registration, on campus residence, socioeconomic level, parental education level, desk, book collection, internet & computer, TV watching time, internet time, sleep time, and exercise time).
Point 8: Statistical analysis. I found no mention of sampling weights. However, the should be used in the analysis?
Response 8: Thank you very much for your valuable advice. More realistic information can be obtained by using sample weight in statistical analysis especially when the distribution of the sample is quite different from the population distribution. However, the China Education Tracking Survey (CEPS) adopts a multi-stage Probability Proportion to Size (PPS) sampling method, and the sampling process was divided into four stages. The characteristic of PPS sampling is that the part with large content in the population has a high probability of being selected, which can improve the representativeness of the sample and reduce the sampling error. Moreover, our research subjects were relatively simple, and the ages were relatively concentrated, so it was easier to obtain a sample that was consistent with the population distribution. Literature review found that many studies using the CEPS data did not use sampling weight in their analysis [1-3]. For the above reasons, we did not perform sample weighting in our analysis.
[1]Shen W. A tangled web: The reciprocal relationship between depression and educational outcomes in China. Soc Sci Res. 2020,85:102353. doi: 10.1016/j.ssresearch.2019.102353.
[2]Fang G, Chan PWK, Kalogeropoulos P. Social Support and Academic Achievement of Chinese Low-Income Children: A Mediation Effect of Academic Resilience. Int J Psychol Res. 2020, 13(1):19-28. doi: 10.21500/20112084.4480.
[3]Yu L, Chen W. The Effect of Boarding on Obesity Among Middle School Students: Evidence From China. Am J Health Promot. 2021,35(2):186-192. doi: 10.1177/0890117120951054.
Point 9: Statistical analysis. The study evaluated more than one outcome (Total score, Chinese score, English score and Math score). Thus, some adjustment for multiple comparisons should be done.
Response 9: Many thanks for your precious suggestions. Differences between the scores of the three subjects were compared using a Generalized Linear Mixed Model with Sidak correction. According to your suggestion, we added a sentence in the Statistical analysis section which as following “The Sidak method was used to correct for multiple testing.” (Line 175-176)
Point 10: 3.3. Distribution of Student Scores, lines 188-190. This point needs further clarifications.
Response 10: Thank you for your careful review. The original scores of the three subjects were available in the CEPS database. Since the full marks of the three subjects varied in different regions, For example, the full score of Chinese in some regions was 100 points, while in other regions it was maybe 120 points. We standardized them using a percentage scoring system, and added them together to obtain a standardized total score. We have revised these sentences to make it clearer as following “Since the full marks of the three subjects varied in different regions, we standardized them using a percentage scoring system, and added them together to obtain a standardized total score, as shown in Table 2.” (Line 220-222)
Point 11: Table 4. Again, since subjects are clustered in countries, schools and classes, the only valid estimates are those obtained from Model 3.
Response 11: Thank you for your precious and helpful suggestions. According to your advice, we combined the Tables 4-7 as Table 4. Meanwhile, to illustrate the whole analysis process of the GLMM models, we keep the results of Model I and Model II as supplementary material (Table S1). Please see Response 7 for the details about the Table 4 and Table S1.
Point 12: Tables 4-7. I would combine these tables in one table by reporting only the estimates obtained by Model 3.
Response 12: Thank you for your precious and helpful suggestions. According to your advice, we combined the Tables 4-7 as Table 4. Meanwhile, to illustrate the whole analysis process of the GLMM models, we keep the results of Model I and Model II as supplementary material (Table S1). Please see Response 7 for the details about the Table 4 and Table S1.
Point 13: Discussion. Line 294. Remove “generally speaking”
Response 13: Thank you for reviewing our manuscript. We have removed “generally speaking” in the paragraph.
Point 14: Discussion. Lines 336-345. The fact that in China high energy diets include foods rich in proteins is of utmost importance in interpreting the results. However, the reader needs to reach the second half of the discussion to understand the reasons of this “unexpected” result. Please mention in the abstract, that the positive association between high-energy diets and academic performance may be the related to the food items included in the high-energy diets consumed by Chinese students.
Response 14: Thank you very much for your constructive and perceptive suggestions. We have added “the positive association between high-energy diets and academic performance may be related to the food items included in the high-energy diets consumed by Chinese students.” to the abstract. (Line 33-35)
Point 15: Lines 383-386. What does it mean that the reporting period for the dietary data was not specified? It is common practice to have an index period when asking a person about their dietary habits. Please clarify.
Response 15: Thanks for reviewing our manuscript. The reporting period for the dietary data was not specified means that the questions about the dietary frequency were not followed by a sentence about time constraints. This is because that the question itself wanted to ask the respondents about the frequency of these foods for a long-term intake, which is similar to the result chosen by the respondent in the psychological test at the first reaction, which should be the best choice that reflect the actual situation. Since the effect of diet on academic performance is usually a long-term effect, we think it was also reasonable not to set a time frame here.
Point 16: Conclusions, lines 397-399. Why they should? The authors found that consuming high-energy diets is associated with better academic performance. Thus, I was wondering why schools and families should advice children against high-energy diets. Please make the conclusions consistent with the results.
Response 16: We deeply appreciate your scholarly review and professional advice. From the available evidence, the effect of a high-energy diet on academic performance is controversial. Most studies showed that lower intakes of fast food were associated with higher academic achievement, but some have linked excessive intake of saturated fatty acids to impaired reading and writing skills. After adjusting for multiple covariates, the results of our study showed that medium high-energy diets were positively associated with academic performance. Therefore, we suggest that students can eat a moderate amount of high-energy food, but avoid overeating. In order to keep the conclusion consistent with the results, we have changed the sentence “Schools and families should pay more effort to reduce consumption of SSBs and meanwhile control the excessive consumption of high-energy diets.” into “Schools and families should pay more effort to reduce consumption of SSBs and meanwhile allowing students to eat moderate amounts of high-energy foods.” (Line 455-456)

Reviewer 2 Report
Dear authors,
The article is well written and brings interesting results. However, some adjustments and additional information should be incorporated into the text for better understanding by the reader. Suggestions are listed below:
Introduction
1) although the rationale of the research is very well founded in the introduction, making clear the importance of carrying out the study, the gaps in the literature, it is important to present, even in a summarized way, the mechanisms that can explain the associations already found in other studies. How can higher consumption of SSB and junk foods reduce academic performance?
Methods
2) Was the questionnaire used in the study applied directly to the students, or were part of the questions asked directly with those responsible? It was in the form of a face-to-face interview or self-completion. Make this information clear in the methods.
3) why not evaluate academic performance considering all the subjects of the school curriculum. Some students may have greater difficulty in exactly the subjects evaluated, which leads to a reduction in their performance in the way it was evaluated, which can also bring a bias to the study. I understand that this consideration must be made in the limitations of the study.
4) the questions asked to assess the consumption of SSB and high-energy diets are very generic and qualitative. The perception of what is frequent for one may be different for another. In addition, there may be cases where consumption is infrequent but when it is consumed, consumption is high. Consider this aspect in the discussion of the results.
5) Was the sample weight considered in the analyses, considering the nature of the sample? If yes, it should be clear in the methods in the analysis section.
6) Was there any question in the questionnaire about cognitive deficit problems, use of medication related to learning, such as medication to improve concentration, etc? this information is also important because school performance depends on numerous factors, in addition to those investigated in this study.
Results
7) It is important to include the 95% confidence intervals in table 1
8) I suggest presenting the frequency of consumption of SSB and high-energy diets presented in Figure 1 separately for boys and girls, as well as the score scores (outcome variable)
9) it was not clear in the text why the authors chose to run separate models for each discipline (mathematics, Chinese and English) and not just the sum of the 3. Are there any results in the literature that support this? For example, any results that showed that frequent consumption of SSB mainly affects performance in exact subjects?
Discussion
10) I suggest that the first paragraph of the discussion brings the main findings of the study, including referring to the models used. In this case, the data in the second paragraph should come in the first
11) I think it is necessary to further explore the personal and family variables that were used in the models and their relationship with school performance.
Author Response
Response to Reviewer 2 Comments
Thank you very much for taking time out of your busy schedule to review our manuscript, and your scholarly review and professional comments have been very helpful to us. We have responded in detail to each of the points raised and these responses are given below.
Comments and Suggestions for Authors
Dear authors,
The article is well written and brings interesting results. However, some adjustments and additional information should be incorporated into the text for better understanding by the reader. Suggestions are listed below:
Introduction
- although the rationale of the research is very well founded in the introduction, making clear the importance of carrying out the study, the gaps in the literature, it is important to present, even in a summarized way, the mechanisms that can explain the associations already found in other studies. How can higher consumption of SSB and junk foods reduce academic performance?
Response 1: Thank you very much for your professional advice. According to your suggestion, we have added the mechanisms that can explain the relationship between SSBs and high-energy diets and academic performance in junior school students that already found in previous studies in the introduction. The following sentences were added in the 3rd paragraph: “The mechanisms that can explain the associations already found in previous studies were mostly about the effects of students’ memory and concentration, which can lead to lower academic performance.” (Line 78-81)
Methods
- Was the questionnaire used in the study applied directly to the students, or were part of the questions asked directly with those responsible? It was in the form of a face-to-face interview or self-completion. Make this information clear in the methods.
Response 2: Thank you very much for your constructive and perceptive suggestions. China Education Panel Survey (CEPS) is maintained by the National Survey Research Center at Renmin University of China and is a nationally representative, longitudinal, social survey, the survey included students, parents, teachers and school leaders, so, questionnaires were designed for students, parents, teachers and school leaders, respectively. Students data were collected by issuing student questionnaires that applied directly to students, and students self-report under the guidance of investigators. According to your suggestion, we have added “Student data were collected by distributing student questionnaires which applied directly to students, and students reported on their own under the guidance of investigators.” in the 2.1. Study Participants and Sample Collection section. (Line 118-120)
- why not evaluate academic performance considering all the subjects of the school curriculum. Some students may have greater difficulty in exactly the subjects evaluated, which leads to a reduction in their performance in the way it was evaluated, which can also bring a bias to the study. I understand that this consideration must be made in the limitations of the study.
Response 3: Thank you for your careful review and kind reminding. Chinese, Math and English are the main courses for junior school students in China, Students of all grades are required to take these three courses. However, other subjects, such as physics and chemistry, are only offered to senior students. The study time of these three courses is longer than that of other subjects, and students spend more energy on them, which can better reflect students' academic performance. So, we only chose Chinese, Math and English to reflect students' academic performance. It does have some limitations to select only the scores of these three courses as the academic performance of students. According to your valuable suggestion, we have added “In addition, the academic performance were only evaluated based on the core subjects of Chinese, Mathematics, and English, and other subjects were not included, which may not fully reflect the students' learning status.” in the limitations of the study. (Line 443-445)
- the questions asked to assess the consumption of SSB and high-energy diets are very generic and qualitative. The perception of what is frequent for one may be different for another. In addition, there may be cases where consumption is infrequent but when it is consumed, consumption is high. Consider this aspect in the discussion of the results.
Response 4: Thank you very much for reading our manuscript carefully and providing us with some keen scientific insights, which are very helpful for us to revise the manuscript and make it better. People's judgment of frequency is subjective, and it would be more objective for participants to make a choice if given the consumption times or ranges in each frequency. Therefore, in future similar studies, we will use more objective and clear methods to judge frequency, such as: seldom(≤1 time/month). From the results, the frequency of intake of SSBs and high-energy diets of the students in this survey is in line with other studies. We include this aspect in the limitations of the manuscript as following: “The consumption frequency was self-reported and students’ judgment of frequency is subjective, so we need to admit the possibility of some reporting bias.” (Line 441-442)
- Was the sample weight considered in the analyses, considering the nature of the sample? If yes, it should be clear in the methods in the analysis section.
Response 5: Thank you very much for your valuable advice. More realistic information can be obtained by using sample weight in statistical analysis especially when the distribution of the sample is quite different from the population distribution. However, the China Education Tracking Survey (CEPS) adopts a multi-stage Probability Proportion to Size (PPS) sampling method, and the sampling process was divided into four stages. The characteristic of PPS sampling is that the part with large content in the population has a high probability of being selected, which can improve the representativeness of the sample and reduce the sampling error. Moreover, our research subjects were relatively simple, and the ages were relatively concentrated, so it was easier to obtain a sample that was consistent with the population distribution. Literature review found that many studies using the CEPS data did not use sampling weight in their analysis [1-3]. For the above reasons, we did not perform sample weighting in our analysis.
[1]Shen W. A tangled web: The reciprocal relationship between depression and educational outcomes in China. Soc Sci Res. 2020,85:102353. doi: 10.1016/j.ssresearch.2019.102353.
[2]Fang G, Chan PWK, Kalogeropoulos P. Social Support and Academic Achievement of Chinese Low-Income Children: A Mediation Effect of Academic Resilience. Int J Psychol Res. 2020, 13(1):19-28. doi: 10.21500/20112084.4480.
[3]Yu L, Chen W. The Effect of Boarding on Obesity Among Middle School Students: Evidence From China. Am J Health Promot. 2021,35(2):186-192. doi: 10.1177/0890117120951054.
- Was there any question in the questionnaire about cognitive deficit problems, use of medication related to learning, such as medication to improve concentration, etc? this information is also important because school performance depends on numerous factors, in addition to those investigated in this study.
Response 6: Thank you very much for your professional advice. This study developed a standardized cognitive test for students, but did not include questions about the use of learning-related medication, such as medication to improve concentration. Academic performance is related to several variables, including unnmodifiable factors such as heredity and gender and modifiable factors such as family environment and personal habits. The purpose of this study was to explore the relationship between SSBs, high-energy diet and academic performance, controlling for student and family characteristics, personal habits, region, schools, and classes. In future similar studies, variables such as learning-related drugs should be taken into account to explore the influence factors of academic performance.
Results
- It is important to include the 95% confidence intervals in table 1
Response 7: Thank you for your careful review and kind reminding. According to your advice, we revised the Table 1. The new Table was shown below.
Table 1. Description of the general participant characteristics
|
|
Variables |
|
N |
Proportion (%) |
95%CI |
|
Student Characteristics |
Gender |
Female |
4444 |
48.0 |
(47.0, 49.1) |
|
|
|
Male |
4807 |
52.0 |
(50.9, 53.0) |
|
|
Region of residence |
Urban |
4418 |
47.8 |
(46.7, 48.8) |
|
|
|
Rural |
4833 |
52.2 |
(51.2, 53.3) |
|
|
Campus residence |
No |
2780 |
30.1 |
(29.1, 31.0) |
|
|
|
Yes |
6471 |
69.9 |
(69.0, 70.9) |
|
Family Characteristics |
Household income |
Low |
1418 |
15.3 |
(14.6, 16.1) |
|
|
|
Medium |
6777 |
73.3 |
(72.3, 74.2) |
|
|
|
High |
1056 |
11.4 |
(10.8, 12.1) |
|
|
Highest level of Parental Education |
Primary or Less |
735 |
7.9 |
(7.4, 8.5) |
|
|
High school |
6657 |
72.0 |
(71.0, 72.9) |
|
|
|
|
College or above |
1859 |
20.1 |
(19.3, 20.9) |
|
|
Independent desk |
Yes |
7270 |
78.6 |
(77.7, 79.4) |
|
|
|
No |
1981 |
21.4 |
(20.6, 22.3) |
|
|
Domestic library |
Lower |
1135 |
12.3 |
(11.6, 12.9) |
|
|
|
Low |
1348 |
14.6 |
(13.9, 15.3) |
|
|
|
Medium |
3547 |
38.3 |
(37.4, 39.3) |
|
|
|
High |
2095 |
22.6 |
(21.8, 23.5) |
|
|
|
Higher |
1126 |
12.2 |
(11.5, 12.8) |
|
|
Internet & Computer |
No |
2459 |
26.6 |
(25.7, 27.5) |
|
|
|
One of them |
6792 |
73.4 |
(72.5, 74.3) |
|
Personal habits |
Screen (TV) |
No |
2940 |
31.8 |
(30.8, 32.7) |
|
|
|
<1h |
2267 |
24.5 |
(23.6, 25.4) |
|
|
|
1-2h |
2182 |
23.6 |
(22.7, 24.5) |
|
|
|
2-3h |
1045 |
11.3 |
(10.7,12.0) |
|
|
|
3-4h |
312 |
3.4 |
(3.0, 3.8) |
|
|
|
>4h |
505 |
5.5 |
(5.0, 5.9) |
|
|
Play online games |
No |
3954 |
42.7 |
(41.7, 43.8) |
|
|
|
<1h |
2044 |
22.1 |
(21.3, 22.9) |
|
|
|
1-2h |
1684 |
18.2 |
(17.4, 19) |
|
|
|
2-3h |
779 |
8.4 |
(7.9, 9.0) |
|
|
|
3-4h |
289 |
3.1 |
(2.8, 3.5) |
|
|
|
>4h |
501 |
5.4 |
(5.0, 5.9) |
|
|
Sleep |
<5h |
61 |
0.7 |
(0.5, 0.8) |
|
|
|
5-9h |
7876 |
85.1 |
(84.4, 85.9) |
|
|
|
>9h |
1314 |
14.2 |
(13.5, 14.9) |
|
|
Sports |
≤60min |
8160 |
88.2 |
(87.5, 88.9) |
|
|
|
60-180min |
967 |
10.5 |
(9.8, 11.1) |
|
Dietary factors |
|
>180min |
124 |
1.3 |
(1.1, 1.6) |
|
SSBs |
Seldom |
3366 |
36.4 |
(35.4, 37.4) |
|
|
|
|
Sometimes |
3896 |
42.1 |
(41.1, 43.1) |
|
|
|
Often |
1989 |
21.5 |
(20.7, 22.3) |
|
|
High-energy diet |
Seldom |
3801 |
41.1 |
(40.1, 42.1) |
|
|
|
Sometimes |
4096 |
44.3 |
(43.3, 45.3) |
|
|
|
Often |
1345 |
14.6 |
(13.9, 15.4) |
- I suggest presenting the frequency of consumption of SSB and high-energy diets presented in Figure 1 separately for boys and girls, as well as the score scores (outcome variable)
Response 8: Thank you very much for your beneficial advice. According to your suggestion, we revised the Figure 1 and Table 2. We also added some description about the Figure 1 as following “There were statistically significant differences in the frequency of SSBs and high-energy diets between genders. Specifically, the proportion of male students who “often” consumed SSBs was higher than that of female students (23.7% vs 19.1%, χ2=30.039, p<0.001), and the proportion of female students who “often” consumed high-energy diets was higher than that of male students (16.0% vs 13.4%, χ2=36.807, p<0.001).” (Line 197-202)
Figure 1. The percent of the consumption frequency of SSBs and high-energy diets reported by male students (A) and female stduents (B).
Table 2. Description of the academic performance
|
Variables |
N |
Male |
Female |
Total |
|
M (P25, P75)a |
M (P25, P75)a |
M (P25, P75)a |
||
|
Standardized total score |
9251 |
192.5(189.7, 195.0) |
220.8 (218.3, 222.5) |
206. 7(153.0, 242.5) |
|
Standardized Chinese score |
9251 |
68.0(67.5, 69.0) |
75.0(75.0, 75.8) |
71. 7(61.3, 79.0) |
|
Standardized Math score |
9251 |
67.0(66.0, 68.7) |
73.0(72.3, 74.0) |
70.0(45.0, 85.4) |
|
Standardized English score |
9251 |
59.0(57.5, 60.0) |
73.3(72.7, 74.5) |
66. 7(43.0, 82. 7) |
a Data were not normally distributed therefore medians and quartiles were used.
- it was not clear in the text why the authors chose to run separate models for each discipline (mathematics, Chinese and English) and not just the sum of the 3. Are there any results in the literature that support this? For example, any results that showed that frequent consumption of SSB mainly affects performance in exact subjects?
Response 9: Thanks to the reviewer for reviewing our manuscript. Chinese, Math and English are three different subjects, which reflect students' academic performance in different dimensions, and the relationship between SSBs and high-energy diets and these three subjects is not completely consistent. For example, a study of primary school students in one province of Canada showed that although excessive intake of saturated fatty acids affected reading and writing proficiency, it did not affect math performance [1].
[1]. Faught EL, Ekwaru JP, Gleddie D, Storey KE, Asbridge M, Veugelers PJ. The Combined Impact of Diet, Physical Activity, Sleep and Screen Time on Academic Achievement: A Prospective Study of Elementary School Students in Nova Scotia, Canada. Int J Behav Nutr Phys Act 2017, 14(1):29.
Discussion
10) I suggest that the first paragraph of the discussion brings the main findings of the study, including referring to the models used. In this case, the data in the second paragraph should come in the first.
Response 10: Thank you very much for your constructive and professional suggestions. According to your advice, we added a new paragraph at the beginning of the discussion to bring the main findings of the study. We deleted the second paragraph and the data integrated into the first paragraph. The first paragraph was as following “In the current study, we found a relatively high proportion of the students in “often” consumption groups of SSBs and high-energy diets. We also found an association between SSBs and high-energy diets and test scores. Our univariate analyses indicated that students who reported higher consumption of SSBs had poorer test scores, while who reported medium consumption of high-energy diet had better scores. GLMM analyses showed that, after controlling for fixed and random effects, both SSBs and high-energy diets were still associated with scores. Specifically, high frequency of SSBs consumption had a negatively statistically significant effect on total scores and all three subjects, while moderate consumption of high−energy diets had a positively statistically significant effect on total scores, Chines scores and English scores. Our study suggests that dietary modification support for junior school students is likely to benefit health and education outcomes.” (Line 324-335)
11) I think it is necessary to further explore the personal and family variables that were used in the models and their relationship with school performance.
Response 11: Thank you very much for your perceptive suggestions. We also think that it’s important to further explore the personal and family variables that were used in the models and their relationship with school performance. However, considering that the aim of the current study was to understand the consumption frequency of SSBs and high-energy diets in junior school students in China and to explore the relationship between SSBs and high-energy diets and academic performance. The personal and family variables were included in the study as fixed effects when GLMM analysis was used, the purpose is to obtain a more realistic and reliable results about the relationship between SSBs and high-energy diets and academic performance. There are many personal and family variables involved in this manuscript, and if there is a lot of discussion, it feels like it will weaken the theme of this manuscript. Therefore, we feel that it is better not to have a discussion about the personal and family variables. But you've given us an excellent idea, and we could consider to further explore the personal and family variables and their relationship with school performance in the future studies.

Round 2
Reviewer 1 Report
Thank you for addressing my comments.
There are only two points that need the attention of the authors:
1) The authors have explained in the reply to my comments why they did not use the sampling weights in the analysis. I believe it is important to provide an explanation also in the manuscript.
2) Please report in the footnotes of Table 4 that the p-values were adjusted for multiple comparisons.
Author Response
Many thanks for reviewing our manuscript, and your scholarly review and professional comments have been very helpful to us. Our responses are listed below and thanks again.
Thank you for addressing my comments.
There are only two points that need the attention of the authors:
- The authors have explained in the reply to my comments why they did not use the sampling weights in the analysis. I believe it is important to provide an explanation also in the manuscript
Response 1: Thank you for your careful review and kind reminding. We have added the explanation why we did not perform the sampling weights in the analysis in the 2.5. Statistical analysis section. The sentence is as following: “Since this study adopted the PPS sampling design, and the subjects were relatively simple and the ages were relatively concentrated, it was easier to obtain samples consistent with the population distribution. Therefore, no sample weighting was performed in our analysis.” (Line 161-164)
- Please report in the footnotes of Table 4 that the p-values were adjusted for multiple comparisons
Response 2: Thank you very much for your valuable advice. We have added “The p-values were adjusted for multiple comparisons.” in the footnotes of Table 4. (Line 282-283)
